# Langevin Autoencoders for Learning Deep Latent Variable Models

**Shohei Taniguchi**
The University of Tokyo
taniguchi@weblab.t.u-tokyo.ac.jp

**Yusuke Iwasawa**
The University of Tokyo
iwasawa@weblab.t.u-tokyo.ac.jp

**Wataru Kumagai**
The University of Tokyo
kumagai@weblab.t.u-tokyo.ac.jp

**Yutaka Matsuo**
The University of Tokyo
matsuo@weblab.t.u-tokyo.ac.jp

## Abstract

Markov chain Monte Carlo (MCMC), such as Langevin dynamics, is valid for approximating intractable distributions. However, its usage is limited in the context of deep latent variable models owing to costly datapoint-wise sampling iterations and slow convergence. This paper proposes the *amortized Langevin dynamics* (ALD), wherein datapoint-wise MCMC iterations are entirely replaced with updates of an encoder that maps observations into latent variables. This amortization enables efficient posterior sampling without datapoint-wise iterations. Despite its efficiency, we prove that ALD is valid as an MCMC algorithm, whose Markov chain has the target posterior as a stationary distribution under mild assumptions. Based on the ALD, we also present a new deep latent variable model named the *Langevin autoencoder* (LAE). Interestingly, the LAE can be implemented by slightly modifying the traditional autoencoder. Using multiple synthetic datasets, we first validate that ALD can properly obtain samples from target posteriors. We also evaluate the LAE on the image generation task, and show that our LAE can outperform existing methods based on variational inference, such as the variational autoencoder, and other MCMC-based methods in terms of the test likelihood.

## 1 Introduction

Variational inference (VI) and Markov chain Monte Carlo (MCMC) are two practical tools to approximate intractable distributions. Recently, VI has been dominantly used in deep latent variable models (DLVMs) to approximate the posterior distribution over the latent variable $\mathbf{z}$ given the observation $\mathbf{x}$, i.e., $p(\mathbf{z} \mid \mathbf{x})$. At the core of the success of VI is the invention of amortized variational inference (AVI) [Kingma and Welling, 2013, Rezende et al., 2014], which replaces optimization of datapoint-wise variational parameters with an encoder that predicts latent variables from observations. The framework of learning DLVMs based on AVI is called the variational autoencoder (VAE), which is widely used in applications [Kingma et al., 2014, An and Cho, 2015, Zhang et al., 2016, Eslami et al., 2018, Kumar et al., 2018]. An important advantage of AVI over traditional VI is that the optimized encoder can be used to infer a latent representation even for new data in test time, expecting its generalization ability. On the other hand, the approximation power of AVI (or VI itself) is limited because it relies on tractable distributions for complex posterior approximations of DLVMs as shown in Figure 1 (a). Although there have been attempts to improve their flexibility (e.g., normalizing flows [Rezende and Mohamed, 2015, Kingma et al., 2016, Van Den Berg et al., 2018, Huang et al., 2018]), such methods typically have architectural constraints (e.g., invertibility in normalizing flows).

36th Conference on Neural Information Processing Systems (NeurIPS 2022).

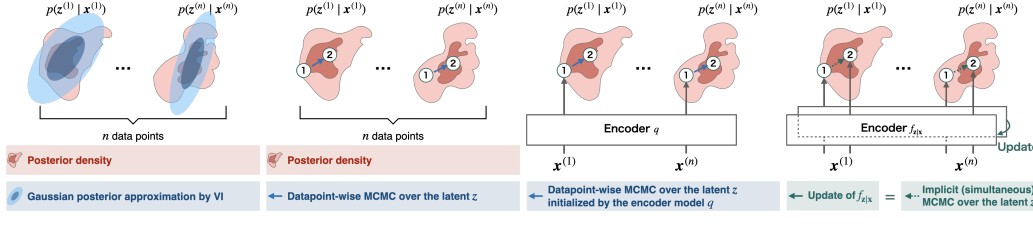

(a) Variational Inference (b) Langevin Dynamics  (c) Hoffman [2017]  (d) ALD (ours)

Figure 1: Comparison between existing approximated inference methods and our amortized Langevin dynamics (ALD). (a) In variational inference, posteriors are approximated using tractable distributions (e.g., Gaussians). (b) In traditional Langevin dynamics (LD), the approximation is performed by running gradient-based sampling iterations directly on the latent space for each datapoint. (c) Hoffman [2017] uses an encoder that maps the observation into the latent variable to initialize traditional LD, but it still relies on datapoint-wise iterations. (d) Our ALD also uses an encoder, but it treats the output of the encoder as a posterior sample, and its update is implicitly performed by updating the encoder.

Compared to VI, MCMC (e.g., Langevin dynamics) can approximate complex distributions by repeating sampling from a Markov chain that has the target distribution as its stationary distribution [Liu and Liu, 2001, Robert et al., 2004, Gilks et al., 1995, Geyer, 1992]. However, despite its high approximation ability, MCMC has received relatively little attention in learning DLVMs. It is because MCMC methods take a long time to converge, making it difficult to be used in the training of DLVMs. When learning DLVMs with MCMC, we need to run time-consuming MCMC iterations for sampling from each posterior per data point, i.e., $p\left(\mathbf{z} \mid \boldsymbol{x}^{(i)}\right)$ $(i = 1, \ldots, n)$, where $n$ is the number of mini-batch data, as shown in Figure 1 (b). Furthermore, we need to re-run the sampling procedure when we obtain new data at test time.

As in VI, there have been some attempts to introduce the concept of amortized inference to MCMC. For example, Hoffman [2017] initializes datapoint-wise MCMC using an encoder that predicts latent variables from observations as shown in Figure 1 (c). However, as they use encoders only for the initialization of MCMC, these methods still rely on datapoint-wise sampling iterations. Not only is it time-consuming, but implementations of such partially amortized methods also tend to be complicated compared to the simplicity of AVI. To make MCMC more suitable for the inference of DLVMs, a more sophisticated framework of amortization is needed.

This paper proposes *amortized Langevin dynamics* (ALD), which entirely replaces datapoint-wise MCMC iterations with updates of an encoder that maps the observation into the latent variable as shown in Figure 1 (d). Since the latent variable depends on the encoder, the updates of the encoder can be regarded as implicit updates of latent variables. By replacing MCMC on the latent space with one on the encoder's parameter space, we can benefit from the encoder's generalization ability. For example, after running a Markov chain of the encoder for some mini-batch data, it is expected that the encoder can map data into high density area in the latent space; hence we can accelerate the convergence of MCMC for other data by initializing it with the encoder. Moreover, despite the simplicity, we can theoretically guarantee that ALD has the true posterior as a stationary distribution under mild assumptions, which is a critical requirement for valid MCMC algorithms.

Using our ALD for sampling from the latent posterior, we derive a novel framework of learning DLVMs, which we refer to as the *Langevin autoencoder* (LAE). Interestingly, the learning algorithm of LAEs can be regarded as a small modification of traditional autoencoders [Hinton and Salakhutdinov, 2006]. In our experiments, we first show that ALD can properly obtain samples from target distributions using toy datasets. Subsequently, we perform numerical experiments of the image generation task using the MNIST, SVHN, CIFAR-10, and CelebA datasets. We demonstrate that our LAE can outperform existing learning methods based on variational inference, such as the VAE, and existing MCMC-based methods in terms of test likelihood.

## 2 Preliminaries

### 2.1 Problem Definition

Consider a probabilistic model with the observation $\mathbf{x}$, the continuous latent variable $\mathbf{z}$, and the model parameter $\boldsymbol{\theta}$ as follows:

$$p\left(\mathbf{x};\boldsymbol{\theta}\right) = \int p\left(\mathbf{x} \mid \boldsymbol{z};\boldsymbol{\theta}\right) p\left(\boldsymbol{z}\right) d\boldsymbol{z}. \tag{1}$$

To learn this latent variable model (LVM) via maximum likelihood in a gradient-based manner, we need to calculate the expectation over the posterior distribution $p\left(\mathbf{z} \mid \mathbf{x};\boldsymbol{\theta}\right)$ as follows:

$$\nabla_{\boldsymbol{\theta}}\mathbb{E}_{\hat{p}_{\text{data}}(\mathbf{x})}\left[\log p\left(\boldsymbol{x};\boldsymbol{\theta}\right)\right] \approx \frac{1}{n}\sum_{i=1}^{n}\mathbb{E}_{p\left(\mathbf{z}^{(i)}\mid\boldsymbol{x}^{(i)};\boldsymbol{\theta}\right)}\left[\nabla_{\boldsymbol{\theta}}\log p\left(\boldsymbol{x}^{(i)},\boldsymbol{z}^{(i)};\boldsymbol{\theta}\right)\right], \tag{2}$$

where $\hat{p}_{\text{data}}$ is the empirical distribution define by the training set, and $\boldsymbol{x}^{(1)},\ldots,\boldsymbol{x}^{(n)}$ are mini-batch data uniformly drawn from $\hat{p}_{\text{data}}$. However, this expectation cannot be calculated in a closed-form because the posterior is intractable. In this paper, we consider to use Monte Carlo approximation by obtaining samples from the posterior per data point.

### 2.2 Langevin Dynamics

Langevin dynamics (LD) [Neal, 2011] is a sampling algorithm based on the following Langevin equation:

$$d\boldsymbol{z} = -\nabla_{\boldsymbol{z}}U\left(\boldsymbol{x},\boldsymbol{z};\boldsymbol{\theta}\right)dt + \sqrt{2\beta^{-1}}dB, \tag{3}$$

where $U$ is a Lipschitz continuous potential function that satisfies an appropriate growth condition, $\beta$ is an inverse temperature parameter, and $B$ is a Brownian motion. This stochastic differential equation (SDE) has $p^{\beta}\left(\boldsymbol{z} \mid \boldsymbol{x};\boldsymbol{\theta}\right) \propto \exp\left(-\beta U\left(\boldsymbol{x},\boldsymbol{z};\boldsymbol{\theta}\right)\right)$ as its stationary distribution. We set $\beta = 1$ and define the potential as follows to obtain the target posterior $p\left(\boldsymbol{z} \mid \boldsymbol{x};\boldsymbol{\theta}\right)$ as its stationary distribution:

$$U\left(\boldsymbol{x},\boldsymbol{z};\boldsymbol{\theta}\right) = -\log p\left(\boldsymbol{x},\boldsymbol{z};\boldsymbol{\theta}\right). \tag{4}$$

We can obtain samples from the posterior by simulating the SDE of Eq. (3) using the Euler–Maruyama method [Kloeden and Platen, 2013] as follows:

$$\boldsymbol{z}_{t+1} \sim q\left(\boldsymbol{z}_{t+1} \mid \boldsymbol{z}_{t}\right), \tag{5}$$

$$q\left(\boldsymbol{z}' \mid \boldsymbol{z}\right) \coloneqq \mathcal{N}\left(\boldsymbol{z}';\boldsymbol{z} - \eta\nabla_{\boldsymbol{z}}U\left(\boldsymbol{x},\boldsymbol{z};\boldsymbol{\theta}\right), 2\eta\boldsymbol{I}\right), \tag{6}$$

where $\eta$ is a step size for discretization. The initial value $\boldsymbol{z}_0$ is typically sampled from the prior $p\left(\boldsymbol{z}\right)$. When the step size is sufficiently small, the samples asymptotically move to the target posterior by repeating this sampling iteration. To remove the discretization error, an additional Metropolis-Hastings (MH) rejection step is often used. In a MH step, we first calculate the acceptance rate $\alpha_t$ as follows:

$$\alpha_t = \min\left\{1, \frac{\exp\left(-U\left(\boldsymbol{x},\boldsymbol{z}_{t+1};\boldsymbol{\theta}\right)\right)q\left(\boldsymbol{z}_t \mid \boldsymbol{z}_{t+1}\right)}{\exp\left(-U\left(\boldsymbol{x},\boldsymbol{z}_t;\boldsymbol{\theta}\right)\right)q\left(\boldsymbol{z}_{t+1} \mid \boldsymbol{z}_t\right)}\right\} \tag{7}$$

The sample $\boldsymbol{z}_{t+1}$ is accepted with probability $\alpha_t$, and rejected with probability $1 - \alpha_t$. If the sample is rejected, we set $\boldsymbol{z}_{t+1} = \boldsymbol{z}_t$. LD can be applied to any posterior inference problems for continuous latent variables, provided the potential energy is differentiable on the latent space. To obtain the posterior samples for all mini-batch data $\boldsymbol{x}^{(1)},\ldots\boldsymbol{x}^{(n)}$, we should perform iterations of Eq. (5) per data point, as shown in Figure 1 (b).

After obtaining the samples, the gradient in Eq. (2) is approximated using the samples as follows:

$$\nabla_{\boldsymbol{\theta}}\mathbb{E}_{\hat{p}_{\text{data}}(\mathbf{x})}\left[\log p\left(\boldsymbol{x};\boldsymbol{\theta}\right)\right] \approx \frac{1}{nT}\sum_{i=1}^{n}\sum_{t=1}^{T}\nabla_{\boldsymbol{\theta}}\log p\left(\boldsymbol{x}^{(i)},\boldsymbol{z}_t^{(i)};\boldsymbol{\theta}\right), \tag{8}$$

The time averaged gradient in Eq. (8) is sometimes substituted for the one calculated with the final sample as follows:

$$\nabla_{\boldsymbol{\theta}}\mathbb{E}_{\hat{p}_{\text{data}}(\mathbf{x})}\left[\log p\left(\boldsymbol{x};\boldsymbol{\theta}\right)\right] \approx \frac{1}{n}\sum_{i=1}^{n}\nabla_{\boldsymbol{\theta}}\log p\left(\boldsymbol{x}^{(i)},\boldsymbol{z}_T^{(i)};\boldsymbol{\theta}\right), \tag{9}$$

**Algorithm 1** Amortized Langevin dynamics

---

$\phi \leftarrow$ Initialize parameters
$\mathbb{Z}^{(1)}, \ldots, \mathbb{Z}^{(n)} \leftarrow \varnothing$                              ▷ Initialize sample sets for all $n$ datapoints.
**repeat**
    $\phi' \sim q\left(\phi' \mid \phi\right) := \mathcal{N}\left(\phi'; \phi - \eta \sum_{i=1}^{n} \nabla_\phi U\left(\boldsymbol{x}^{(i)}, \boldsymbol{z}^{(i)} = f_{\mathbf{z}|\mathbf{x}}\left(\boldsymbol{x}^{(i)}; \phi\right)\right), 2\eta \boldsymbol{I}\right)$
    $\phi \leftarrow \phi'$ with probability $\min\left\{1, \frac{\exp(-V(\phi'))q(\phi|\phi')}{\exp(-V(\phi))q(\phi'|\phi)}\right\}$.                              ▷ MH rejection step.
    $\mathbb{Z}^{(1)}, \ldots, \mathbb{Z}^{(n)} \leftarrow \mathbb{Z}^{(1)} \cup \left\{f_{\mathbf{z}|\mathbf{x}}\left(\boldsymbol{x}^{(1)}; \phi\right)\right\}, \ldots, \mathbb{Z}^{(N)} \cup \left\{f_{\mathbf{z}|\mathbf{x}}\left(\boldsymbol{x}^{(n)}; \phi\right)\right\}$      ▷ Add samples.
**until** convergence of parameters
**return** $\mathbb{Z}^{(1)}, \ldots, \mathbb{Z}^{(n)}$

---

However, in these traditional approaches, we need to run MCMC iterations from random initialization every time after updating the model parameter $\boldsymbol{\theta}$. In addition, we also need to run it from stratch to perform inference for new data in test time. This inefficiency hinders the practical use of MCMC for learning DLVMs. A naive approach to mitigate this problem is to initialize MCMC with a persistent sample [Tieleman, 2008, Han et al., 2017] that is the final value of the previous Markov chain. However, this approach is also inefficient especially when the training set is large, because we need to store persistent samples for all training examples.

To alleviate the inefficiency of LD for LVMs, Hoffman [2017] has proposed to use a stochastic encoder $q\left(\mathbf{z} \mid \mathbf{x}\right)$ to initialize the datapoint-wise MCMC as shown in Figure 1 (c). The encoder is typically defined as a Gaussian distribution as in the VAE:

$$q\left(\boldsymbol{z} \mid \mathbf{x}; \phi\right) = \mathcal{N}\left(\boldsymbol{z}; \mu\left(\boldsymbol{x}; \phi\right), \mathrm{diag}\left(\sigma^2\left(\boldsymbol{x}; \phi\right)\right)\right), \tag{10}$$

where $\mu$ and $\sigma^2$ are mappings from the observation space into the latent space. In the training, LD iterations of Eq. (5) are initialized using a sample from the distribution of Eq. (10), and the model parameter is updated using the stochastic gradient in Eq. (9). Along with it, the encoder is trained via maximization of the evidence lower bound as in VAEs:

$$\mathcal{L}\left(\phi\right) = \mathbb{E}_{q(\mathbf{z}|\mathbf{x})\hat{p}(\mathbf{x})}\left[\log \frac{p\left(\boldsymbol{x}, \boldsymbol{z}; \boldsymbol{\theta}\right)}{q\left(\mathbf{z} \mid \mathbf{x}; \phi\right)}\right]. \tag{11}$$

Although initializing LD with the encoder can speed up the convergence, this method still relies on datapoint-wise MCMC iterations. Moreover, the encoder has to be trained using the different objective, which also makes its implementation complicated. In Section 3, we demonstrate a method that entirely remove the datapoint-wise iterations by amortization.

## 3 Method

### 3.1 Amortized Langevin Dynamics

As an alternative to the direct simulation of latent dynamics of Eq. (3), we define a deterministic encoder $f_{\mathbf{z}|\mathbf{x}}$, which maps the observation into the latent variable, and consider an SDE over its parameter $\phi$ as follows:

$$d\phi = -\nabla_\phi V\left(\phi\right) dt + \sqrt{2}dB, \tag{12}$$

$$V\left(\phi\right) := \sum_{i=1}^{n} U\left(\boldsymbol{x}^{(i)}, f_{\mathbf{z}|\mathbf{x}}\left(\boldsymbol{x}^{(i)}; \phi\right); \boldsymbol{\theta}\right). \tag{13}$$

Because the function $f_{\mathbf{z}|\mathbf{x}}$ outputs the latent variable, the stochastic dynamics on the parameter space induces dynamics on the latent space. The main idea of our amortized Langevin dynamics (ALD) is to regard the transition on this induced dynamics as a sampling procedure for the posterior distributions, as shown in Figure 1 (d). We can use the Euler–Maruyama method to simulate Eq. (12) with discretization:

$$\phi_{t+1} \sim q\left(\phi_{t+1} \mid \phi_t\right), \tag{14}$$

$$q\left(\phi' \mid \phi\right) := \mathcal{N}\left(\phi'; \phi - \eta \nabla_\phi V\left(\phi\right), 2\eta \boldsymbol{I}\right). \tag{15}$$

As in the traditional LD, the discretization error can be removed by adding a MH rejection step, calculating the acceptance rate as follows:

$$\alpha_t = \min \left\{ 1, \frac{\exp\left(-V\left(\phi_{t+1}\right)\right) q\left(\phi_t \mid \phi_{t+1}\right)}{\exp\left(-V\left(\phi_t\right)\right) q\left(\phi_{t+1} \mid \phi_t\right)} \right\} \tag{16}$$

Through the iterations, the posterior sampling is implicitly performed by collecting outputs of the encoder for each data point as described in Algorithm 1. Note that $\mathbb{Z}^{(i)}$ denotes a set of samples of the posterior for the $i$-th data (i.e., $p\left(\mathbf{z} \mid \boldsymbol{x}^{(i)}\right)$) obtained using ALD.

If this implicit update iteration has the posterior as its stationary distribution, this sampling procedure is valid as an MCMC algorithm. To obtain the stationary distribution over the latent variables, we first derive the stationary distribution over the parameter $\phi$ of Eq. (12), then transform it into the latent space by considering the change of random variable by the encoder $f_{\mathbf{z} \mid \mathbf{x}}$. Based on this strategy, we derive the following theorem:

**Theorem 1.** *Let $q\left(\boldsymbol{Z} \mid \boldsymbol{X}\right) \coloneqq \prod_{i=1}^n q\left(\boldsymbol{z}^{(i)} \mid \boldsymbol{x}^{(i)}\right)$ be a stationary distribution over the latent variables that is induced by Eq. (12). When the mapping $f_{\mathbf{z} \mid \mathbf{x}}$ meets the following conditions, the stationary distribution satisfies $q\left(\boldsymbol{Z} \mid \boldsymbol{X}\right) \propto \exp\left(-\sum_{i=1}^n U\left(\boldsymbol{x}^{(i)}, \boldsymbol{z}^{(i)}\right)\right)$.*

1. *The mapping has the form of $f_{\mathbf{z} \mid \mathbf{x}}\left(\boldsymbol{x}; \boldsymbol{\Phi}\right) = \boldsymbol{\Phi} g\left(\boldsymbol{x}\right)$, where $\boldsymbol{\Phi}$ is a $d_{\mathbf{z}} \times d$ matrix, $g$ is a mapping from $\mathbb{R}^{d_{\mathbf{x}}}$ to $\mathbb{R}^d$, and $d_{\mathbf{x}}$, $d_{\mathbf{z}}$ and $d$ are the dimensionalities of $\boldsymbol{x}$, $\boldsymbol{z}$ and $g\left(\boldsymbol{x}\right)$ respectively.*

2. *The rank of $\boldsymbol{G}$ is $n$, where $\boldsymbol{G}$ is a matrix with $g\left(\boldsymbol{x}^{(i)}\right)$ in row $\boldsymbol{G}_{i,:}$.*

See the appendix for the solid proof. Theorem 1 suggests that samples obtained by ALD asymptotically converge to the true posterior when we construct the encoder $f_{\mathbf{z} \mid \mathbf{x}}$ with an appropriate form. Practically, we can implement such a function using a neural network whose parameters are fixed except for the last linear layer. In this implementation, the last linear layer takes a role of the parameter $\boldsymbol{\Phi}$, and the preceding feature extractor takes a role of the function $g$.

To meet the second condition, the dimensionality of the last linear layer should be larger than the batch size $n$. It is relatively easy to meet this condition because the batch size is about 1,000 at most in practice.

## 3.2 Remarks

**Remark 1: ALD completely removes datapoint-wise iterations**.

Because ALD treats the encoder's outputs themselves as posterior samples, we no longer need to run time-consuming iterations of datapoint-wise MCMC, whereas existing methods, such as Hoffman [2017] only use the encoder for initialization of datapoint-wise MCMC.

**Remark 2: ALD is valid as an MCMC algorithm**.

Although the sampling procedure of ALD is quite simple, ALD has the true posterior as its stationary distribution under mild assumptions, which guarantee that ALD is valid as an MCMC algorithm. Basically, we can meet the assumptions by using a sufficiently wide neural network for the encoder. In addition, it is worth mentioning that the traditional LD can be seen as a special case of ALD where $g\left(\boldsymbol{x}^{(i)}\right) = \text{one-hot}\left(i\right)$. In that case, $\boldsymbol{z}^{(i)} = f_{\mathbf{z} \mid \mathbf{x}}\left(\boldsymbol{x}^{(i)}; \boldsymbol{\Phi}\right)$ corresponds to the $i$-th column of $\boldsymbol{\Phi}$, which is equivalent to running MCMC on the latent space independently for each data point.

**Remark 3: The encoder may accelerate the convergence of MCMC**.

After running ALD iterations of the encoder for some mini-batch data, it is expected that the encoder can map data into high density area in the latent space. Therefore, the encoder can accelerate the convergence of MCMC for other data by initializing it with the encoder. This characteristics is useful not only in the training of DLVMs to efficiently estimate the gradient of the model parameters but also at test time to infer the latent representation for new test data.

---

**Algorithm 2** Langevin Autoencoders

---

1: $\boldsymbol{\theta}, \boldsymbol{\Phi}, \boldsymbol{\psi} \leftarrow$ Initialize parameters
2: **repeat**
3:     $\boldsymbol{x}^{(1)}, \ldots, \boldsymbol{x}^{(n)} \sim \hat{p}(\mathbf{x})$            ▷ Sample a minibatch of $n$ examples from the training data.
4:     **for** $t = 0, \ldots, T-1$ **do**                    ▷ Run ALD iterations.
5:         $V_t = -\sum_{i=1}^{n} \log p\left(\boldsymbol{x}^{(i)}, \boldsymbol{z}^{(i)} = \boldsymbol{\Phi}g\left(\boldsymbol{x}^{(i)}; \boldsymbol{\psi}\right); \boldsymbol{\theta}\right)$
6:         $\boldsymbol{\Phi}' \sim q\left(\boldsymbol{\Phi}' \mid \boldsymbol{\Phi}\right) \coloneqq \mathcal{N}\left(\boldsymbol{\Phi}'; \boldsymbol{\Phi} - \eta\nabla_{\boldsymbol{\Phi}}V_t, 2\eta\boldsymbol{I}\right)$
7:         $V_t' = -\sum_{i=1}^{n} \log p\left(\boldsymbol{x}^{(i)}, \boldsymbol{z}^{(i)} = \boldsymbol{\Phi}'g\left(\boldsymbol{x}^{(i)}; \boldsymbol{\psi}\right); \boldsymbol{\theta}\right)$
8:         $\boldsymbol{\Phi} \leftarrow \boldsymbol{\Phi}'$ with probability $\min\left\{1, \frac{\exp(-V_t')q(\boldsymbol{\Phi}|\boldsymbol{\Phi}')}{\exp(-V_t)q(\boldsymbol{\Phi}'|\boldsymbol{\Phi})}\right\}$.        ▷ MH rejection step.
9:     **end for**
10:     $V_T = -\sum_{i=1}^{n} \log p\left(\boldsymbol{x}^{(i)}, \boldsymbol{z}^{(i)} = \boldsymbol{\Phi}g\left(\boldsymbol{x}^{(i)}; \boldsymbol{\psi}\right); \boldsymbol{\theta}\right)$
11:     $\boldsymbol{\theta} \leftarrow \boldsymbol{\theta} - \eta\nabla_{\boldsymbol{\theta}}\frac{1}{T}\sum_{t=1}^{T} V_t$                ▷ Update the decoder.
12:     $\boldsymbol{\psi} \leftarrow \boldsymbol{\psi} - \eta\nabla_{\boldsymbol{\psi}}\frac{1}{T}\sum_{t=1}^{T} V_t$                ▷ Update the encoder.
13: **until** convergence of parameters
14: **return** $\boldsymbol{\theta}, \boldsymbol{\Phi}, \boldsymbol{\psi}$

---

### 3.3 Langevin Autoencoder

Based on ALD, we propose a novel framework of learning deep latent variable models called the *Langevin autoencoder* (LAE). Algorithm 2 is a summary of the LAE's training procedure. First, we prepare an encoder defined by $f_{\mathbf{z}|\mathbf{x}}(\boldsymbol{x}; \boldsymbol{\Phi}, \boldsymbol{\psi}) \coloneqq \boldsymbol{\Phi}g(\boldsymbol{x}; \boldsymbol{\psi})$. The feature extractor $g$ is typically implemented using a deep neural network. Before each update of the model parameter $\boldsymbol{\theta}$, the encoder's final linear layer $\boldsymbol{\Phi}$ is updated using ALD for $T$ times, and the gradient of $\boldsymbol{\theta}$ in Eq. (8) is calculated using the time average of $V_t$. Along with the update of the model parameter, the encoder's feature extractor $\boldsymbol{\psi}$ is also updated using the gradient so that the encoder can map data into high density area of the posteriors. Although we describe the parameter update using a simple stochastic gradient ascent, it can be substituted for more sophisticated optimization methods, such as Adam [DP and Ba, 2015].

It can be seen that the algorithm is very similar to the one for traditional deterministic autoencoders [Hinton and Salakhutdinov, 2006]. In particular, when we skip the ALD iterations in lines 4 to 9 and the encoder's final linear layer $\boldsymbol{\Phi}$ is also trained via gradient ascent, the algorithm is identical to the training of autoencoders regularized by the latent prior $p(\boldsymbol{z})$. In that case, the encoder tends to shrink to the maximum a posteriori (MAP) estimate rather than the posterior; hence the gradient of the model parameter $\boldsymbol{\theta}$ would be a strongly biased estimate of the marginal log-likelihood gradient. Therefore, the additional ALD iterations can be interpreted as the modification to reduce the bias of the traditional autoencoder by updating the encoder's parameters along with the noise-injected gradient.

## 4 Related Works

**Amortized inference** is well-investigated in the context of variational inference. It is often referred to as *amortized variational inference* (AVI) [Rezende and Mohamed, 2015, Shu et al., 2018]. The basic idea of AVI is to replace the optimization of the datapoint-wise variational parameters with the optimization of shared parameters across all datapoints by introducing an encoder that predicts latent variables from observations. The AVI is commonly used in generative models [Kingma and Welling, 2013], semi-supervised learning [Kingma et al., 2014], anomaly detection [An and Cho, 2015], machine translation [Zhang et al., 2016], and neural rendering [Eslami et al., 2018, Kumar et al., 2018]. However, in the MCMC literature, there are few works on such amortization. Han et al. [2017] use traditional LD to obtain samples from posteriors for the training of deep latent variable models. Such Langevin-based algorithms for deep latent variable models are known as *alternating back-propagation* (ABP) and are widely applied in several fields [Xie et al., 2019, Zhang et al., 2020, Xing et al., 2018, Zhu et al., 2019]. However, ABP requires datapoint-wise Langevin

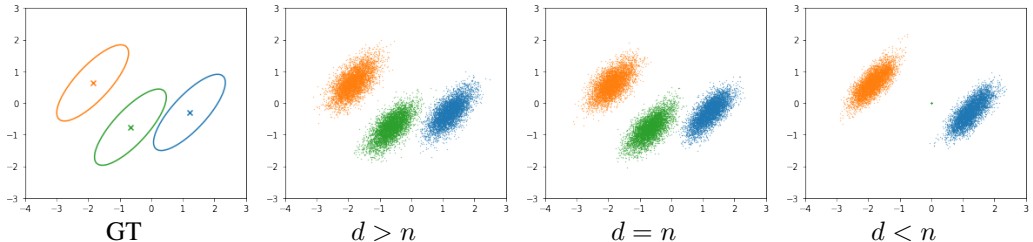

Figure 2: Effects of changing the encoder's capacity in the experiment of bivariate Gaussian examples. $d$ denotes the dimensionality of the last linear layer's input.

iterations, causing slow convergence. Moreover, when we perform inference for new data in test time, ABP requires MCMC iterations from randomly initialized samples again. Li et al. [2017] and Hoffman [2017] propose to use a VAE-like encoder to initialize MCMC, and Salimans et al. [2015] also proposes to combine VAE-based inference and MCMC by interpreting each MCMC step as an auxiliary variable. However, they only amortize the initialization cost in MCMC by using an encoder; hence, they still rely on datapoint-wise MCMC iterations.

**Autoencoders** (AEs) [Hinton and Salakhutdinov, 2006] are a special case of LAEs, wherein the ALD iterations are omitted and a uniform distribution is used as the latent prior $p(z)$. When a different distribution is used as the latent prior as regularization, it is known as sparse autoencoders (SAEs) [Ng et al., 2011]. As described in the previous section, the encoder of the traditional AE tends to converge to MAP estimates of the latent posterior. Therefore, the gradient of the decoder's parameter is biased as the gradient estimate of the marginal log-likelihood. Our LAE can modify this bias by adding the ALD iterations before each parameter update, making it valid as training of a generative model.

**Variational Autoencoders** (VAEs) are based on AVI, wherein an encoder is defined as a variational distribution $q(\mathbf{z} \mid \mathbf{x}; \phi)$ using a neural network. Its parameter $\phi$ is optimized by maximizing the evidence lower bound (ELBO), i.e., $\mathbb{E}_{q(\mathbf{z}|\boldsymbol{x};\phi)}\left[\log \frac{\exp(-U(\boldsymbol{x}, \boldsymbol{z}))}{q(\boldsymbol{z}|\boldsymbol{x};\phi)}\right]$. Interestingly, there is a contrast between VAEs and LAEs relative to when stochastic noise is used in posterior inference. In VAEs, noise is used to sample from the variational distribution in calculating the potential $U$, i.e., in the *forward* calculation. However, in LAEs, noise is used for the parameter update along with the gradient calculation $\nabla_\phi U$, i.e., in the *backward* calculation. This contrast characterizes their two different approaches to approximate posteriors: the optimization-based approach of VAEs and the sampling-based approach of LAEs. The advantage of LAEs over VAEs is that LAEs can flexibly approximate complex posteriors by obtaining samples, whereas VAEs' approximation ability is limited by choice of variational distribution $q(\mathbf{z} \mid \mathbf{x}; \phi)$ because it requires a tractable density. Although there are several considerations in the improvement of the approximation flexibility, these methods typically have architectural constraints (e.g., invertibility and ease of Jacobian calculation in normalizing flows [Rezende and Mohamed, 2015, Kingma et al., 2016, Van Den Berg et al., 2018, Huang et al., 2018, Titsias and Ruiz, 2019]), or they incur more computational costs (e.g., MCMC sampling for the reverse conditional distribution in unbiased implicit variational inference [Titsias and Ruiz, 2019]).

## 5 Experiment

In our experiment, we first test our ALD algorithm on toy examples to investigate its behavior, then we show the results of its application to the training of deep generative models for image datasets.

### 5.1 Toy Examples

We perform numerical simulation using toy examples to demonstrate that our ALD can properly obtain samples from target distributions. First, we use a LVM where the posterior density can be derived in a closed-form. We initially generate three synthetic data $\boldsymbol{x}_1, \boldsymbol{x}_2, \boldsymbol{x}_3$, where each $\boldsymbol{x}_i$ is sampled from a bivariate Gaussian distribution as follows:

$$p(z) = \mathcal{N}(\boldsymbol{\mu}_\mathbf{z}, \boldsymbol{\Sigma}_\mathbf{z}), \quad p(\boldsymbol{x} \mid z) = \mathcal{N}(\boldsymbol{z}, \boldsymbol{\Sigma}_\mathbf{x}). \tag{17}$$

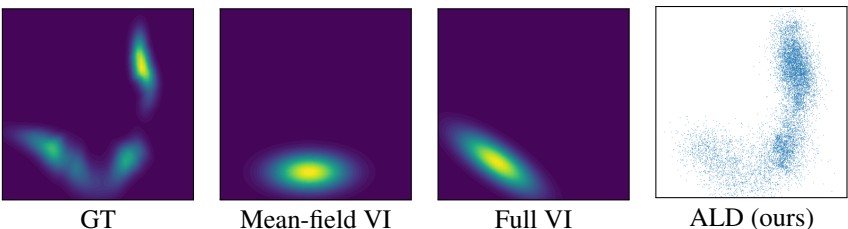

| GT | Mean-field VI | Full VI | ALD (ours) |

Figure 3: Visualizations of a ground truth posterior (left), an approximation by VI (center), and samples by ALD (right) in the neural likelihood example.

In this case, we can calculate the exact posterior as follows:

$$p\left(\boldsymbol{z} \mid \boldsymbol{x}\right) = \mathcal{N}\left(\boldsymbol{\mu}_{\mathbf{z}|\mathbf{x}}, \boldsymbol{\Sigma}_{\mathbf{z}|\mathbf{x}}\right), \tag{18}$$

$$\boldsymbol{\mu}_{\mathbf{z}|\mathbf{x}} = \boldsymbol{\Sigma}_{\mathbf{z}|\mathbf{x}}\left(\boldsymbol{\Sigma}_{\mathbf{z}}^{-1}\boldsymbol{\mu}_{\mathbf{z}} + \boldsymbol{\Sigma}_{\mathbf{x}}^{-1}\boldsymbol{x}\right), \quad \boldsymbol{\Sigma}_{\mathbf{z}|\mathbf{x}} = \left(\boldsymbol{\Sigma}_{\mathbf{z}}^{-1} + \boldsymbol{\Sigma}_{\mathbf{x}}^{-1}\right)^{-1}. \tag{19}$$

In this experiment, we set $\boldsymbol{\mu}_{\mathbf{z}} = \begin{bmatrix} 0 \\ 0 \end{bmatrix}$, $\boldsymbol{\Sigma}_{\mathbf{z}} = \begin{bmatrix} 1 & 0 \\ 0 & 1 \end{bmatrix}$, and $\boldsymbol{\Sigma}_{\mathbf{x}} = \begin{bmatrix} 0.7 & 0.6 \\ 0.7 & 0.8 \end{bmatrix}$. We simulate our ALD algorithm for this setting to obtain samples from the posterior. We use a neural network of three fully connected layers of 128 units with ReLU activation for the encoder $f_{\mathbf{z}|\mathbf{x}}$; setting the step size to $4 \times 10^{-4}$, and update the parameters for 3,000 steps. We omit the first 1,000 samples as burn-in steps and use the remaining 2,000 samples for qualitative evaluation. As we described in Section 3.1, we need to make the last linear layer of the encoder have sufficiently large dimensions to guarantee the convergence to the true posterior. To empirically demonstrate this theoretical finding, we change the dimensionality of the last linear layer of the encoder from 2 $(< n)$ to 128 $(\gg n)$.

The results are summarized in Figure 2. It can be observed that the sample quality is good when the dimensionality of the last linear layer is equal to or greater than the number of data points (i.e., $d \geq n$). When the dimensionality is smaller than the number of data points, the samples for some data points shrink to a small area, while good samples are obtained for the remaining data points.

In addition to the simple conjugate Gaussian example, we experiment with a complex posterior, wherein the likelihood is defined with a randomly initialized neural network. For comparison, we also implement the variational inference (VI), in which the posterior is approximated with a Gaussian distribution. Figure 3 shows a typical example, which characterizes the difference between VI and ALD. The mean-filed VI and the full VI use Gaussians with diagonal and full covariance matrices for variational distributions, respectively. The advantage of our ALD over VI is the flexibility of posterior approximation. VI methods typically approximate posteriors using variational distributions, which have tractable density functions. Hence, their approximation power is limited by the choice of variational distribution family, and they often fail to approximate such complex posteriors. In particular, the mean-field VI, which is widely used for learning DLVMs, cannot capture the correlation between dimensions due to the constraint of the variational distribution. The full VI mitigate the inflexibility, but it still cannot capture the multimodality of the true posterior. Moreover, the full VI requires computational costs proportional to the square of the dimension of the latent variable. On the other hand, ALD can capture such posteriors well. The results in other examples are summarized in the appendix.

## 5.2 Image Generation

To demonstrate the applicability of our LAE to the generative model training, we experiment on image generation tasks using MNIST, SVHN, CIFAR10, and CelebA datasets. Note that our goal here is not to provide the state-of-the-art results on image generation benchmarks but to verify the effectiveness of our ALD as a method of approximate inference in deep latent variable models. For this aim, we compare our LAE with some baseline methods, as shown in Table 1. The VAE [Kingma and Welling, 2013] is one of the most popular deep latent variable models, in which the posterior distribution is approximated using the VI. The VAE-flow is an extension of VAE, in which the flexibility of VI is improved using normalizing flows [Rezende and Mohamed, 2015]. In addition to VI-based methods, we use Hoffman [2017] as a baseline method based on Langevin dynamics (LD). As described in Section 2.2, Hoffman [2017] uses a VAE-like encoder to initialize LD, and the encoder is trained by

Table 1: Quantitative results of the image generation for MNIST, SVHN, CIFAR-10, and CelebA. We report the mean and standard deviation of the negative evidence lower bound per data dimension in three different seeds. Lower is better.

|  | MNIST | SVHN | CIFAR-10 | CelebA |
| --- | --- | --- | --- | --- |
| VAE | $1.189 \pm 0.002$ | $4.442 \pm 0.003$ | $4.820 \pm 0.005$ | $4.671 \pm 0.001$ |
| VAE-flow | $1.183 \pm 0.001$ | $4.454 \pm 0.016$ | $4.828 \pm 0.005$ | $4.667 \pm 0.005$ |
| Hoffman [2017] | $1.189 \pm 0.002$ | $4.440 \pm 0.007$ | $4.831 \pm 0.005$ | $4.662 \pm 0.011$ |
| LAE (ours) | $\mathbf{1.177} \pm 0.001$ | $\mathbf{4.412} \pm 0.002$ | $\mathbf{4.773} \pm 0.003$ | $\mathbf{4.636} \pm 0.003$ |

maximizing the evidence lower bound. We use the same fully connected deep neural networks for the model construction of all methods. We set the number of ALD iterations of the LAE to 2, i.e, $T = 2$ in Algorithm 2. Please refer to the appendix for more implementation details.

For evaluation, since the marginal log-likelihood for test data is not tractable, we substitute its evidence lower bound (ELBO) for it using a proposal distribution $q$ as follows:

$$\log p\left(\boldsymbol{x};\boldsymbol{\theta}\right) \geq \mathbb{E}_{q(\mathbf{z})}\left[\log \frac{p\left(\boldsymbol{x},\boldsymbol{z};\boldsymbol{\theta}\right)}{q\left(\boldsymbol{z}\right)}\right] \tag{20}$$

For the baseline methods, their stochastic encoders are used for the proposal distribution, i.e., $q\left(\boldsymbol{z}\right) \coloneqq q\left(\boldsymbol{z} \mid \boldsymbol{x};\boldsymbol{\phi}\right)$. For our LAE, we use a Gaussian distribution whose mean parameter is defined by its encoder, i.e., $q\left(\boldsymbol{z}\right) \coloneqq \mathcal{N}\left(\boldsymbol{z};\boldsymbol{\Phi}g\left(\boldsymbol{x};\boldsymbol{\psi}\right),\ \sigma^2\boldsymbol{I}\right)$. We set $\sigma = 0.05$ in the experiment.

The results are summarized in Table 1. Note that the negative ELBO is shown in the table, so lower values indicate better results. It can be observed that the LAE consistently outperforms baseline methods, demonstrating that accurate posterior approximation by ALD leads to better results in the training of DLVMs. On training speed, we observe that our LAE is 2.24 and 1.88 times slower than VAE and VAE-flow respectively. This is natural because VAE and VAE-flow do not require MCMC steps. Hoffman [2017] and our LAE are almost identical in terms of training speed.

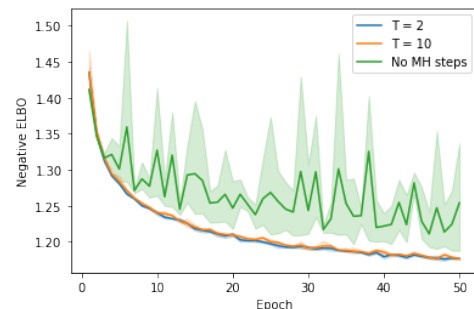

Figure 4: Learning curves comparison.

We also investigate the effect of the MH rejection step and the number of ALD iterations, i.e., $T$ in Algorithm 2, using MNIST dataset. Figure 4 shows a comparison of the learning curves of the negative ELBO for MNIST's test set. It can be seen that the number of ALD iterations has little effect on performance as long as the number of steps is at least 2. In addition, we observe that the MH rejection step is important to stabilize the training.

## 6    Conclusion

This paper proposed amortized Langevin dynamics (ALD), an efficient MCMC method for deep latent variable models (DLVMs). ALD amortizes the cost of datapoint-wise iterations by using an encoder that predict the latent variable from the observation. We showed that our ALD algorithm could accurately approximate posteriors with both theoretical and empirical studies. Using ALD, we derived a novel scheme of deep generative models called the *Langevin autoencoder* (LAE). We demonstrated that our LAE performs better than VI-based methods, such as the variational autoencoder, and existing LD-based methods in terms of the marginal log-likelihood for test sets.

This study will be the first step to further work on encoder-based MCMC methods for latent variable models. For instance, developing algorithms based on more sophisticated Hamiltonian Monte Carlo methods is an exciting direction of future work.

One of the limitations of MCMC-based learning algorithms is the difficulty in choosing the number of MCMC iterations. To reduce the bias of the gradient estimate, we need to run the iterations for many times, but there are few clues as to how many MCMC iterations are sufficient in advance.

Recently, a method to remove the bias of MCMC with couplings is proposed by Jacob et al. [2020], and it may help to overcome this limitation of MCMC-based learning algorithm in the future work. Another limitation of our LAE is that there is a constraint on the structure of the encoder as described in Theorem 1. Although the constraint is relatively minor, it may be problematic when applying our method to modern DLVMs that have a hierarchical structure in the latent variables (e.g., Vahdat and Kautz [2020] and Child [2020]).

From a broader perspective, developing deep generative models that can synthesize realistic images could cause a negative impact, such as abuse of Deepfake technology. We must consider the negative aspects and take measures for them.

## Acknowledgments and Disclosure of Funding

This work was supported by JSPS KAKENHI Grant Number JP21J22342 and the Mohammed bin Salman Center for Future Science and Technology for Saudi-Japan Vision 2030 at The University of Tokyo (MbSC2030). Computational resources of AI Bridging Cloud Infrastructure (ABCI) provided by National Institute of Advanced Industrial Science and Technology (AIST) were used for the experiments.

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
