# A Proof of Theorem 1

First, we prepare some lemmas.

**Lemma 2.** *Let $h : \mathbb{R}^{d_{\mathbf{z}} \times d} \to \mathbb{R}^{d_{\mathbf{z}} \times n}$ be a linear map defined by $h\left(\mathbf{\Phi}\right) := \mathbf{\Phi} G$. When the rank of $G$ is $n$, there exists an orthogonal linear map $\tau : \mathbb{R}^{d_{\mathbf{z}} \times d} \to \mathbb{R}^{d_{\mathbf{z}} \times d}$ such that $\tau\left(\mathbf{\Phi}\right) = [\tilde{\mathbf{\Phi}}^{(1)}, \tilde{\mathbf{\Phi}}^{(2)}]$ satisfies $\ker \tilde{h} = \mathrm{span}\left[\mathbf{0}_{d_{\mathbf{z}} \times n}, \tilde{\mathbf{\Phi}}^{(2)}\right]$, where $\tilde{h} := h \circ \tau^{-1}$, $\tilde{\mathbf{\Phi}}^{(1)} := \left[\tilde{\phi}_1, \ldots, \tilde{\phi}_n\right]$, $\tilde{\mathbf{\Phi}}^{(2)} := \left[\tilde{\phi}_{n+1}, \ldots \tilde{\phi}_d\right]$ and $\tilde{\phi}_i \in \mathbb{R}^{d_{\mathbf{z}}}$ for $i = 1, \ldots, d$.*

*Proof.* The singular value decomposition of $G$ is represented by $TG\tilde{T}^\top = \begin{bmatrix} \mathbf{\Lambda} \\ \mathbf{0}_{(d-n \times n)} \end{bmatrix}$, where $\mathbf{\Lambda}$ is a $n \times n$ diagonal matrix $\mathbf{\Lambda} = \mathrm{diag}\left(\lambda_1, \ldots, \lambda_n\right)$, and $T$ and $\tilde{T}$ are orthogonal matrices. Since the rank of $G$ is $n$, $\lambda_1, \ldots, \lambda_n$ are non-zero. When we set $\tau\left(\mathbf{\Phi}\right) := \mathbf{\Phi}T^\top$, we obtain

$$\tilde{h}\left(\tilde{\mathbf{\Phi}}\right) := h\left(\tau^{-1}\left(\tilde{\mathbf{\Phi}}\right)\right)$$
$$= \tilde{\mathbf{\Phi}}TG = \tilde{\mathbf{\Phi}}\left(TG\tilde{T}^\top\right)\tilde{T}$$
$$= \tilde{\mathbf{\Phi}}\begin{bmatrix} \mathbf{\Lambda} \\ \mathbf{0}_{(d-n \times n)} \end{bmatrix}\tilde{T}. \tag{21}$$

From the above equation, $\ker \tilde{h} = \mathrm{span}\left[\mathbf{0}_{d' \times n}, \tilde{\mathbf{\Phi}}^{(2)}\right]$ holds. □

**Lemma 3.** *For $\tilde{\mathbf{\Phi}} \in \mathbb{R}^{d_{\mathbf{z}} \times d}$, let $\mathbf{\Phi}$ satisfy:*

$$\tilde{\mathbf{\Phi}} = \left[\tilde{\mathbf{\Phi}}^{(1)}, \tilde{\mathbf{\Phi}}^{(2)}\right] = \tau\left(\mathbf{\Phi}\right).$$

*A map $\tilde{h}^{(1)} : \mathbb{R}^{d_{\mathbf{z}} \times n} \to \mathbb{R}^{d_{\mathbf{z}} \times n}$ defined by $\tilde{h}^{(1)}\left(\tilde{\mathbf{\Phi}}^{(1)}\right) := \tilde{h}\left(\left[\tilde{\mathbf{\Phi}}^{(1)}, \mathbf{0}\right]\right)$ satisfies $\mathbf{\Phi}G = \tilde{h}^{(1)}\left(\tilde{\mathbf{\Phi}}^{(1)}\right)$ and is linear isomorphic.*

*Proof.* From the definition, we have

$$\mathbf{\Phi}G = \tau^{-1}\left(\tilde{\mathbf{\Phi}}\right)G = \tilde{\mathbf{\Phi}}TG$$
$$= \tilde{\mathbf{\Phi}}\left(TG\tilde{T}^\top\right)\tilde{T}$$
$$= \left[\tilde{\mathbf{\Phi}}^{(1)}, \tilde{\mathbf{\Phi}}^{(2)}\right]\begin{bmatrix} \mathbf{\Lambda} \\ \mathbf{0}_{(d-n \times n)} \end{bmatrix}\tilde{T}$$
$$= \left[\tilde{\mathbf{\Phi}}^{(1)}, \mathbf{0}\right]\begin{bmatrix} \mathbf{\Lambda} \\ \mathbf{0}_{(d-n \times n)} \end{bmatrix}\tilde{T}$$
$$= \tau^{-1}\left(\left[\tilde{\mathbf{\Phi}}^{(1)}, \mathbf{0}\right]\right)G$$
$$= h \circ \tau^{-1}\left(\left[\tilde{\mathbf{\Phi}}^{(1)}, \mathbf{0}\right]\right) = \tilde{h}\left(\left[\tilde{\mathbf{\Phi}}^{(1)}, \mathbf{0}\right]\right)$$
$$= \tilde{h}^{(1)}\left(\tilde{\mathbf{\Phi}}^{(1)}\right).$$

By the definition, $\tilde{h}^{(1)}$ is linear. Here, $\tilde{h}^{(1)}$ is injective, since $\ker \tilde{h} = \mathrm{span}\left[\mathbf{0}_{d' \times n}, \tilde{\mathbf{\Phi}}^{(2)}\right]$, and hence, $\dim\left(\mathrm{Im}\,\tilde{h}^{(1)}\right) \geq d_{\mathbf{z}} \times n$. Since $\mathrm{Im}\,\tilde{h}^{(1)} \subset \mathbb{R}^{d_{\mathbf{z}} \times n}$, $\tilde{h}^{(1)}$ is surjective. □

**Lemma 4.** *For $V : \mathbb{R}^D \ni \phi \mapsto V\left(\mathbf{\Phi}\right) = U\left(X, f_{\mathbf{z}|\mathbf{x}}\left(X; \mathbf{\Phi}\right)\right) := \sum_{i=1}^n U\left(x^{(i)}, f_{\mathbf{z}|\mathbf{x}}\left(x^{(i)}; \mathbf{\Phi}\right)\right) \in \mathbb{R}$, $\tilde{\mathbf{\Phi}} = \left[\tilde{\mathbf{\Phi}}^{(1)}, \tilde{\mathbf{\Phi}}^{(2)}\right] := \tau\left(\mathbf{\Phi}\right)$, $\tilde{V} := V \circ \tau^{-1}$ and $\tilde{V}^{(1)}\left(\tilde{\mathbf{\Phi}}^{(1)}\right) := \tilde{V}\left(\left[\tilde{\mathbf{\Phi}}^{(1)}, \mathbf{0}_{d_{\mathbf{z}} \times (d-n)}\right]\right)$, Eq. (12) is equivalent to*

$$d\tilde{\mathbf{\Phi}}^{(1)} = -\nabla_{\tilde{\mathbf{\Phi}}^{(1)}}\tilde{V}^{(1)}\left(\tilde{\mathbf{\Phi}}^{(1)}\right)dt + \sqrt{2}dB, \tag{22}$$
$$d\tilde{\mathbf{\Phi}}^{(2)} = \sqrt{2}dB. \tag{23}$$

*Proof.* By direct calculation, we obtain

$$\tilde{V}\left(\left[\tilde{\boldsymbol{\Phi}}^{(1)},\tilde{\boldsymbol{\Phi}}^{(2)}\right]\right) \tag{24}$$

$$= V \circ \tau^{-1}\left(\left[\tilde{\boldsymbol{\Phi}}^{(1)},\tilde{\boldsymbol{\Phi}}^{(2)}\right]\right)$$

$$= U\left(\boldsymbol{X}, f_{\mathbf{z}|\mathbf{x}}\left(\boldsymbol{X};\tau^{-1}\left(\left[\tilde{\boldsymbol{\Phi}}^{(1)},\tilde{\boldsymbol{\Phi}}^{(2)}\right]\right)\right)\right)$$

$$= U\left(\boldsymbol{X}, h\left(\tau^{-1}\left(\left[\tilde{\boldsymbol{\Phi}}^{(1)},\tilde{\boldsymbol{\Phi}}^{(2)}\right]\right)\right)\right)$$

$$= U\left(\boldsymbol{X}, h\circ\tau^{-1}\left(\left[\tilde{\boldsymbol{\Phi}}^{(1)},\mathbf{0}\right]\right)+h\circ\tau^{-1}\left(\left[\mathbf{0},\tilde{\boldsymbol{\Phi}}^{(2)}\right]\right)\right)$$

$$= U\left(\boldsymbol{X}, h\left(\tau^{-1}\left(\left[\tilde{\boldsymbol{\Phi}}^{(1)},\mathbf{0}\right]\right)\right)\right)$$

$$= U\left(\boldsymbol{X}, f_{\mathbf{z}|\mathbf{x}}\left(\boldsymbol{X};\tau^{-1}\left(\left[\tilde{\boldsymbol{\Phi}}^{(1)},\mathbf{0}\right]\right)\right)\right)$$

$$= V\circ\tau^{-1}\left(\left[\tilde{\boldsymbol{\Phi}}^{(1)},\mathbf{0}\right]\right)$$

$$= \tilde{V}\left(\left[\tilde{\boldsymbol{\Phi}}^{(1)},\mathbf{0}\right]\right). \tag{25}$$

Then, the following equivalence holds:

$$d\boldsymbol{\Phi} = -\nabla_{\boldsymbol{\Phi}}V\left(\boldsymbol{\Phi}\right)dt + \sqrt{2}dB,$$

$$\Leftrightarrow d\tau^{-1}\left(\boldsymbol{\Phi}\right) = -\tau^{\top}\left(\nabla_{\tilde{\boldsymbol{\Phi}}}\tilde{V}\left(\tilde{\boldsymbol{\Phi}}\right)\right)dt + \sqrt{2}dB$$

$$\Leftrightarrow d\tilde{\boldsymbol{\Phi}} = -\tau\circ\tau^{\top}\left(\nabla_{\tilde{\boldsymbol{\Phi}}}\tilde{V}\left(\tilde{\boldsymbol{\Phi}}\right)\right)dt + \sqrt{2}d\tau\left(B\right) \tag{26}$$

$$= -\nabla_{\tilde{\boldsymbol{\Phi}}}\tilde{V}\left(\tilde{\boldsymbol{\Phi}}\right)dt + \sqrt{2}dB,$$

where we used $\tau\circ\tau^{\top} = \text{id}$ because $\tau$ is orthogonal. From Eq. (25), the dynamics in Eq. (26) is equivalent to Eq. (22) and Eq. (23). $\qquad\square$

In the following, we prove Theorem 1 using the above lemmas.

Because the latent variables $\boldsymbol{Z} := \tilde{h}^{(1)}\left(\tilde{\boldsymbol{\Phi}}^{(1)}\right)$ are independent of $\tilde{\boldsymbol{\Phi}}^{(2)}$, the stationary distribution $q\left(\boldsymbol{Z} \mid \boldsymbol{X}\right)$ is given by $\left(\tilde{h}^{(1)}\right)_{\#}\left(p_*^{(1)}\right)\left(\boldsymbol{Z}\right)$, which is the pushforward measure of the probability distribution $p^{(1)}\left(\tilde{\boldsymbol{\Phi}}\right)$ by $\tilde{h}^{(1)}$. Then, we have

$$q\left(\boldsymbol{Z} \mid \boldsymbol{X}\right)$$

$$= \left(\tilde{h}^{(1)}\right)_{\#}\left(p_*^{(1)}\right)\left(\boldsymbol{Z}\right)$$

$$= p^{(1)}\left(\left(\tilde{h}^{(1)}\right)^{-1}\left(\boldsymbol{Z}\right)\right)\left|\det\frac{d(\tilde{h}^{(1)})^{-1}}{d\boldsymbol{Z}}\right|$$

$$= p^{(1)}\left(\left(\tilde{h}^{(1)}\right)^{-1}\left(\boldsymbol{Z}\right)\right)\left|\det\frac{d\tilde{h}^{(1)}}{d\tilde{\boldsymbol{\Phi}}^{(1)}}\right|^{-1}$$

$$= p^{(1)}\left(\left(\tilde{h}^{(1)}\right)^{-1}\left(\boldsymbol{Z}\right)\right)\times\left|\det\tilde{h}^{(1)}\right|^{-1}$$

$$\propto \exp\left(-\tilde{V}\left(\left(\tilde{h}^{(1)}\right)^{-1}\left(\boldsymbol{Z}\right)\right)\right)$$

$$= \exp\left(-V\left(\tau^{-1}\left(\left[\left(\tilde{h}^{(1)}\right)^{-1}\left(\boldsymbol{Z}\right),\mathbf{0}\right]\right)\right)\right)$$

$$= \exp\left(-U\left(\boldsymbol{X},\tau^{-1}\left(\left[\left(\tilde{h}^{(1)}\right)^{-1}\left(\boldsymbol{Z}\right),\mathbf{0}\right]\right)\boldsymbol{G}\right)\right)$$

$$= \exp\left(-U\left(\boldsymbol{X},\boldsymbol{Z}\right)\right),$$

| GT | VI (diag.) | VI (full) | ALD (hist.) | ALD (sample) |
| --- | --- | --- | --- | --- |

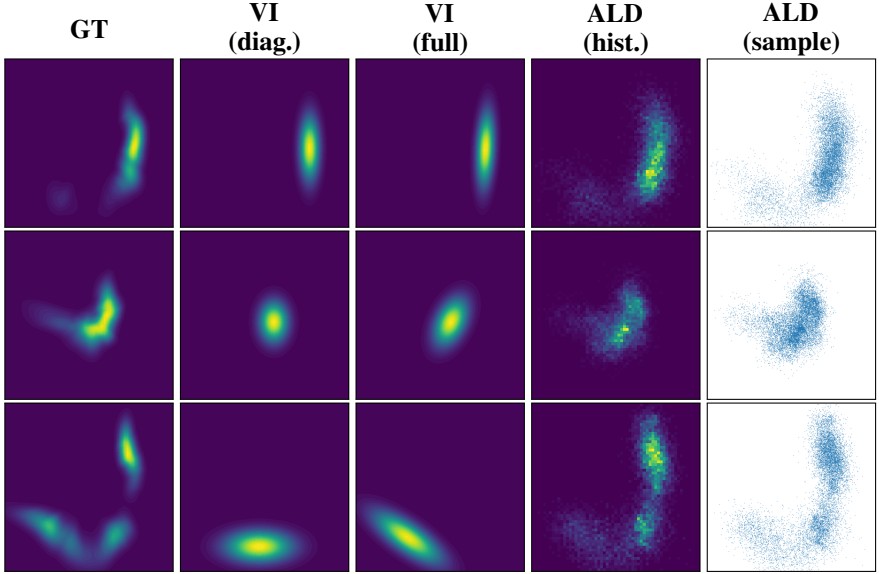

Figure 5: Neural likelihood experiments.

where we used that $\frac{d\tilde{h}^{(1)}}{d\tilde{\boldsymbol{\Phi}}^{(1)}} = \tilde{h}^{(1)}$ because of the linearity of $\tilde{h}^{(1)}$ and is constant with respect to $\boldsymbol{Z}$. The last equation is derived as follows. From Lemma 3, $\boldsymbol{\Phi G} = \tilde{h}^{(1)}\left(\tilde{\boldsymbol{\Phi}}^{(1)}\right)$ holds when $\tilde{\boldsymbol{\Phi}} = \left[\tilde{\boldsymbol{\Phi}}^{(1)}, \tilde{\boldsymbol{\Phi}}^{(2)}\right] = \tau\left(\boldsymbol{\Phi}\right)$. Thus, when $\boldsymbol{\Phi} = \tau^{-1}\left(\left[\tilde{\boldsymbol{\Phi}}^{(1)}, \boldsymbol{0}\right]\right)$, we obtain $\tilde{h}^{(1)}\left(\tilde{\boldsymbol{\Phi}}^{(1)}\right) = \boldsymbol{\Phi G} = \tau^{-1}\left(\left[\tilde{\boldsymbol{\Phi}}^{(1)}, \boldsymbol{0}\right]\right)\boldsymbol{G}$. In particular, for $\tilde{\boldsymbol{\Phi}}^{(1)} = \left(\tilde{h}^{(1)}\right)^{-1}\left(\boldsymbol{Z}\right)$, we have

$$\boldsymbol{Z} = \tilde{h}^{(1)}\left(\left(\tilde{h}^{(1)}\right)^{-1}\left(\boldsymbol{Z}\right)\right)$$
$$= \tau^{-1}\left(\left[\left(\tilde{h}^{(1)}\right)^{-1}\left(\boldsymbol{Z}\right), \boldsymbol{0}\right]\right)\boldsymbol{G}.$$

$\square$

# B  Experimental Settings

## B.1  Neural likelihood example

We perform an experiment with a complex posterior, wherein the likelihood is defined with a randomly initialized neural network $f_\theta$. Particularly, we parameterize $f_\theta$ by four fully-connected layers of 128 units with ReLU activation and two dimensional outputs like $p\left(\mathbf{x} \mid \boldsymbol{z}\right) = \mathcal{N}\left(f_\theta\left(\boldsymbol{z}\right), \sigma_x^2 I\right)$. We initialize the weight and bias parameters with $\mathcal{N}\left(0, 0.2I\right)$ and $\mathcal{N}\left(0, 0.1I\right)$, respectively. In addition, we set the observation variance $\sigma_x$ to 0.25. We used the same neural network architecture for the encoder $f_\phi$. Other settings are same as the conjugate Gaussian experiment described in Section 5.1.

The results are shown in Figure 5. The left three columns show the density visualizations of the ground truth or approximation posteriors of VI methods; the right two columns show the visualizations of 2D histograms and samples obtained using ALD. For VI method, we use two different models. One uses diagonal Gaussians, i.e., $\mathcal{N}\left(\mu\left(\boldsymbol{x}; \phi\right), \text{diag}\left(\sigma^2\left(\boldsymbol{x}; \phi\right)\right)\right)$, for the variational distribution, and the oher uses Gaussians with full covariance $\mathcal{N}\left(\mu\left(\boldsymbol{x}; \phi\right), \Sigma\left(\boldsymbol{x}; \phi\right)\right)$. From the density visualization of GT, the true posterior is multimodal and skewed; this leads to the failure of the Gaussian VI methods notwithstanding considering covariance. In contrast, the samples of ALD accurately capture such a complex distribution, because ALD does not need to assume any tractable distributions for approximating the true posteriors. The samples of ALD capture well the multimodal and skewed posterior, while Gaussian VI methods fail it even when considering covariance.

## B.2 Image Generation

For the image generation experiments, we use a standard Gaussian disrtibution $\mathcal{N}(z; 0, I)$ for the latent prior. The latent dimensionality is set to $8$ for MNIST, $16$ for SVHN, and $32$ for CIFAR-10 and CelebA. The raw images, which take the values in $\{0, 1, \ldots, 255\}$, are scaled into the range from $-1$ to $1$ via preprocessing. Because the values of the preprocessed images are not continuous in a precise sense due to the quantization, it is not desirable to use continuous distributions, e.g., Gaussians, for the likelihood function $p(x \mid z; \theta)$. Therefore, we define the likelihood using a discretized logistic distribution [Salimans et al., 2017] as follows:

$$
\begin{aligned}
p(x \mid z; \theta) &= \prod_i^{d_x} \int_{a_i}^{b_i} \text{Logistic}(\tilde{x}_i; \mu_i, s)\, d\tilde{x}_i, \\
&= \prod_i^{d_x} \left( \sigma\left( \frac{b_i - \mu_i}{s} \right) - \sigma\left( \frac{a_i - \mu_i}{s} \right) \right), \\
a_i &= \begin{cases} -\infty & x = -1 \\ x_i - \frac{1}{255} & \text{otherwise} \end{cases}, \\
b_i &= \begin{cases} \infty & x = 1 \\ x_i + \frac{1}{255} & \text{otherwise} \end{cases},
\end{aligned}
\tag{27}
$$

where $\mu \coloneqq f_{x|z}(z; \theta)$, $f_{x|z} : \mathbb{R}^{d_z} \to \mathbb{R}^{d_x}$. Logistic $(\cdot; \mu, s)$ is the density function of a logistic distribution with the location parameter $\mu$ and the scale parameter $s$, and $\sigma$ is the logistic sigmoid function. We use a neural network with four fully-connected layers for the decoder function $f_{x|z}$. The number of hidden units are set to $1{,}024$, and ReLU is used for the activation function. Before each activation, we apply the layer normalization [Ba et al., 2016] to stabilize training. The scale parameter $s$ is also optimized in the training. Because it has a constraint of $s > 0$, we parameterize $s = \zeta(b)^{-1/2}$, where $\zeta$ is the softplus function, and treat $b$ as a learnable parameter instead. When the model has sufficiently high expressive power, $b$ may diverge to infinity [Rezende and Viola, 2018], so we add a regularization term of $(b + 2\zeta(-b))/m$ to the loss function, where $m$ is the number of training examples. This regularization corresponds to assuming a standard logistic distribution Logistic $(b; 0, 1)$ for the prior distribution of $b$. We optimize the models using stochastic gradient ascent with the learning rate of $1 \times 10^{-4}$ and the batch size of $100$. We run two steps of ALD iterations, i.e., $T = 2$ in Algorithm 2, and the step size $\eta$ is set to $1 \times 10^{-4}$. We use the same experimental settings for the baseline models. We run the training iterations for $50$ epochs for MNIST, SVHN, CIFAR-10 and $20$ epochs for CelebA. The implementation is available at `https://github.com/iShohei220/LAE`.

## B.3 Datasets

All the dataset we use in the experiment is public for non-commercial research purposes. MNIST [LeCun et al., 1998], SVHN [Netzer et al., 2011], CIFAR-10 [Krizhevsky et al., 2009], CelebA [Liu et al., 2015] are available at `http://yann.lecun.com/exdb/mnist/`, `http://ufldl.stanford.edu/housenumbers`, `https://www.cs.toronto.edu/~kriz/cifar.html`, `http://mmlab.ie.cuhk.edu.hk/projects/CelebA.html`, and `https://github.com/tkarras/progressive_growing_of_gans`, respectively. The images of CelebA are resized to $32 \times 32$ in the experiment. We use the default settings of data splits for all datasets.

## B.4 Computational Resources

We run all the experiments on AI Bridging Cloud Infrastructure (ABCI), which is a large-scale computing infrastructure provided by National Institute of Advanced Industrial Science and Technology (AIST). The experiments are performed on Computing Nodes (V) of ABCI, each of which has four NVIDIA V100 GPU accelerators, two Intel Xeon Gold 6148, one NVMe SSD, 384GiB memory, two InfiniBand EDR ports (100Gbps each). Please see `https://abci.ai` for more details.

Table 2: Effects of the number of MCMC iterations of Hoffman [2017]. We report the mean and standard deviation of the negative evidence lower bound per data dimension in three different seeds. Lower is better.

|  |  | MNIST | SVHN | CIFAR-10 | CelebA |
|---|---|---|---|---|---|
| Hoffman [2017] | ($T = 0$) | $1.189 \pm 0.002$ | $4.442 \pm 0.003$ | $4.820 \pm 0.005$ | $4.671 \pm 0.001$ |
| Hoffman [2017] | ($T = 2$) | $1.189 \pm 0.002$ | $4.440 \pm 0.007$ | $4.831 \pm 0.005$ | $4.662 \pm 0.011$ |
| Hoffman [2017] | ($T = 10$) | $1.188 \pm 0.001$ | $4.437 \pm 0.009$ | $4.832 \pm 0.006$ | $4.664 \pm 0.004$ |
| LAE | ($T = 2$) | $\mathbf{1.177} \pm 0.001$ | $\mathbf{4.412} \pm 0.002$ | $\mathbf{4.773} \pm 0.003$ | $\mathbf{4.636} \pm 0.003$ |

## C  Additional Experiments

In the main result in Section 5, we fix the number of MCMC iterations (i.e., $T$) for the model of Hoffman [2017]. In this additional experiment, we further investigate the effect of $T$ by changing it from 0 to 10. Note that when $T = 0$, the model is identical to the normal VAE. The result is shown in Table 2. It can be seen that the effect is relatively small, and our LAE (with $T = 2$) shows better performance than all cases.