# OpenReview forum: "Langevin Autoencoders for Learning Deep Latent Variable Models"
_NeurIPS.cc/2022/Conference — NeurIPS 2022 Accept_

### Official Review · Reviewer_753q · 2022-07-10

**Rating:** 7
**Confidence:** 4
**Soundness:** 4 excellent
**Presentation:** 4 excellent
**Contribution:** 4 excellent

**Summary:**

This paper presents Amortized Langevin Dynamics (ALD), a scalable MCMC algorithm that can be used to sample from high dimensional deep latent variable models.

Unlike competing MCMC algorithms that need to rely on per-datapoint iterations, ALD can sample from the correct posterior distribution by performing updates to an encoder. This gives rise to Langevin Autoencoders, a family of deep LVMs that relies on ALD.

Experiments on a number of benchmark datasets show that Langevin Autoencoders outperform competing models while maintaining scalability.







**Questions:**

See comments in the "weaknesses" section.

**Limitations:**

Yes

**Strengths And Weaknesses:**

STRENGTHS

Most applications of deep LVMs use variational methods to approximate the intractable posterior distribution.
Variational methods are however inherently biased, since they try to approximate a complex posterior distribution with distributions from a typically simple variational family (e.g. Gaussian).

This is not an issue in sample-based MCMC methods, but they do not scale for high dimensional deep LVMs trained on large datasets, since they require expensive per-datapoint iterations.

This paper provides a novel solution to this, that relies on applying amortization ideas to MCMC methods.
The idea of obtaining samples from the posterior distribution by updating an encoder that data points to latent variables is very interesting to me, as it preserves the scalability of variational methods while not making any strong assumption on the approximate posterior distribution.
Also, theoretical and empirical results show that ALD matches the correct stationary posterior distribution even for the complex posterior distributions that are typical of DLVMs.

I believe this paper can have an impact in the neurips community, since it can inspire a new research direction on enoder-based MCMC methods that use amortization and avoid datapoint-wise iterations.


WEAKNESSES

There are some experimental improvements that I believe could further increase the impact of this paper.

1. You state that the goal of the experiments is to show that ALD works, and not that of achieving SOTA results, which is ok. However, for this type of paper that presents a method that can be readily applied to any deep LVM, trying it out on a more SOTA architecture would have made the results much more compelling. The fact that you have not done it makes me wonder if there are some complications to applying it to more complex architectures.

2. How can one monitor the convergence of the MCMC chain? Which convergence diagnostics have you used?

3. Did you ever get in situations where the MH acceptance rate was too low/high?

4. What are the training times of the different models considered in the experiments?


 Minor comments:

* typo in title: variabel -> variable

* line 29. The issue is not that the distribution is tractable as it is implied in the sentence. The issue is that the posterior for these types of models is complex, so the tractable distributions we use are not a good approximation.

* line 41: "in test time" -> "at test time"

---

> ### Author Response · Authors · 2022-08-02
> **Response to Reviewer 753q**
>
> Thank you for your insightful feedback.
> We respond to your comment to address your concerns.
>
>
> **Applicability to modern deep LVMs**
>
> Basically, our ALD can be applied to any deep LVMs, but there is a constraint that the encoder takes the form of $f_{\mathrm{z} \mid \mathrm{x}} \left( x ; \Phi \right) = \Phi g \left( x \right) $ as described in Theorem 1.
> Although this is a relatively minor constraint, it may be a little problematic when applying to modern DLVMs with hierarchical latent variables (e.g., [1, 2]) because in such cases the latent variables tend to have very large dimensions in total, and as a result, the dimension of $\Phi$ also gets large.
> In the revised version, we have added a statement about this architectural constraint of the LAE's encoder to clarify this point in Section 6.
>
>
> **How to check the convergence of MCMC**
>
> Following Hoffman (2017), we simply treat the number of MCMC steps as a hyperparameter, and monitor the ELBO performance.
>
>
> **Training speed comparison**
>
> We observed that VAE and VAE-flow were 2.24 and 1.88 times faster than our LAE respectively.
> This is natural because VAE and VAE-flow do not require MCMC steps.
> The training speed of the Hoffman (2017) model and our LAE was almost identical when we ran the same number of MCMC iterations.
> We have also added this result to the revised version in Section 5.2.
>
>
> **Minor comments**
>
> > line 29. The issue is not that the distribution is tractable as it is implied in the sentence. The issue is that the posterior for these types of models is complex, so the tractable distributions we use are not a good approximation.
>
> Thank you for concrete feedback. We have revised the sentence to clarify this point.
>
>
> We would be glad to respond to any further questions and comments that you may have.
>
> Thanks.
>
>
> **References**
>
> [1] Vahdat, Arash, and Jan Kautz. "NVAE: A deep hierarchical variational autoencoder." Advances in Neural Information Processing Systems 33 (2020): 19667-19679.
>
> [2] Child, Rewon. "Very Deep VAEs Generalize Autoregressive Models and Can Outperform Them on Images." International Conference on Learning Representations. 2020.

---

> > ### Comment · Reviewer_753q · 2022-08-05
> > **Confirm score**
> >
> > Thanks for your feedback and the updated paper, I confirm my accept score.

---

### Official Review · Reviewer_ZVTr · 2022-07-13

**Rating:** 6
**Confidence:** 4
**Soundness:** 3 good
**Presentation:** 3 good
**Contribution:** 2 fair

**Summary:**

The paper introduces amorized Langevin dynamics (ALD), a method for initializing Langevin dynamics in the function parameter space that makes it efficient to perform posterior sampling. Based on this method, the paper presents the Langevin autoencoder (LAE), a deep latent variable model that is easy to implement and has competitive performance compared to similar existing methods.

**Questions:**

- In Hoffman (2017) the encoder parameters are updated via gradient descent on the ELBO. However in this work the gradients are taken w.r.t. $V$ instead. I did not find derivations for this gradient update. How is this gradient update justified? What is the objective function in this case?
- What do you think could be the reason behind the efficiency of performing MCMC in the parameter space instead of the data space?

**Limitations:**

The main limitation of the method seems to be the requirement that the size of the last linear encoder layer needs to be larger than the batch size. This is addressed in the experimental section 5.1.

**Strengths And Weaknesses:**

Strengths:
- The preliminaries and the core method are explained clearly. The drawbacks of existing MCMC and variational inference methods motivate the main ideas very well.
- The algorithm of LAE is presented very clearly (Algorithm 2).
- The idea of performing Langevin sampling in encoder parameter space seems quite novel and interesting. Proof is given on why this MCMC in the encoder parameter space will converge to the true data posterior.

Weaknesses:
- Although it doesn't get in the way of understanding the key ideas, the writing can certainly be improved at a couple of places. Examples include "a more straightforward and sophisticated framework" (line 48), "replacing MCMC on the latent space into the encoder's parameter space" (line 52) and "we substitue it for its evidence lower bound" (line 277, I believe it should be the other way around). Should be easy to fix.
- I find the claim that ALD "completely removes datapoint-wise iterations" slightly misleading, since it does so trivially by turning datapoint-wise iterations into parameter space iterations. In essence it still relies on the encoder (up to the second to last layer) to initialize the MCMC in a high-density region, which is still pretty similar to Hoffman (2017). Maybe it can be shown that this method can have the same or better performance with less MCMC iterations than Hoffman, at which point it would be interesting to discuss why moving to parameter space gives such an advantage.
- Without providing details on hyperparameter search (which can influence results greatly), the experimental results might not be sufficient for concluding that the LAE "outperforms" the existing methods. As mentioned in the last point, I think it would be really beneficial to show how the performance of the Hoffman (2017) model is influenced by the number of MCMC steps used.

---

> ### Author Response · Authors · 2022-08-02
> **Response to Reviewer ZVTr**
>
> Thank you for your insightful feedback. We answer your comment to address your concern.
>
> **Writing errors**
>
> Thank you for your detailed comments on our writing. We have modified the sentences that you pointed out in the revised version, so please check it.
>
> **On comparison to the Hoffman (2017) model**
>
> Based on your concerns about the experimental results, we have added the results of how the number of MCMC steps influences the performance of the Hoffman (2017) model in Appendix C of the revised version.
> We observe that the performance of the Hoffman (2017) model is slightly improved for some datasets by increasing the number of MCMC steps, but our LAE still outperforms it even with fewer MCMC steps.
>
> **Objective of LAE's encoder**
>
> As you mentioned, the LAE's encoder is trained using the gradient w.r.t. $V$ except for the last linear layer.
> This means that the encoder is trained so that it can map data into the high density area of the posterior, because it tries to maximize $\log p \left( x, z = \Phi g \left( x ; \psi \right) \right)$.
> More formally, the objective for the encoder's parameter $\psi$ is $\mathbb{E}_{q \left( \Phi \right)} \left[ V \right]$, where $q \left( \Phi \right)$ is the empirical distribution of $\Phi$ obtained via ALD.
>
> **Why is MCMC on the parameter space efficient?**
>
> When we run MCMC directly on the latent space, the iterations are performed independently for posteriors of each data.
> On the other hand, when performing it on the parameter space, each parameter is responsible for posterior sampling for all data; hence, the parameters are updated using the correlation between data points, which may effectively accelerate the mixing of MCMC.
> In fact, the traditional LD can be interpreted as a special case of ALD where the encoder’s feature extractor $g$ returns one-hot vectors of the data index (i.e., $g \left( x^{(i)} \right) = \textrm{one-hot} \left( i \right)$).
> In that case, $z^{(i)} = f \left( x^{(i)} ; \Phi \right)$ corresponds to the $i$-th column of $\Phi$, which is equivalent to running MCMC on the latent space independently for each data point.
> We have added this explanation to Remark 2 in Section 3.2 in the revised version.
>
> We would be glad to respond to any further questions and comments that you may have.
>
> Thanks.

---

> > ### Comment · Area_Chair_QVro · 2022-08-09
> > **Reviewer: Please reply to author feedback**
> >
> > Your AC

---

### Official Review · Reviewer_xhuG · 2022-07-14

**Rating:** 5
**Confidence:** 3
**Soundness:** 3 good
**Presentation:** 4 excellent
**Contribution:** 3 good

**Summary:**

The paper deals with performing Langevin dynamics as a form of sampling, only using an amortisation mechanism. This reduces the computational overhead of performing the sampling. The method is then verified on synthetic and generated data sets


**Questions:**

You have used EM numerical solver, which has a lof of well defined numerical stability criterion, for example : Higham, D.J., 2001. An algorithmic introduction to numerical simulation of stochastic differential equations. SIAM review, 43(3), pp.525-546. Did you check any of them ?

What is really fascinating about Langevin dynamics is that for a certain class you derive a shadow lemma which means you are tractable towards your “real” underlying distribution for example : https://arxiv.org/abs/1107.2967
Can this analysis somehow improve the theorem on G


**Limitations:**

Yes

**Strengths And Weaknesses:**

The paper is very clearly written and cleanly presented. It is worth mentioning that the authors also did a very good job introducing other related works (albeit a bit late in the text at section 4).
More so I think The main theorem is an important one. The idea of unfing the generated Markov chains with respect to the data set is novel.
In top of that using amortisation is a fantastic idea for a Markovian process since the lack of time correlation can make this efficient much like Mori Zwanzig operators untangle non markovian processes
That being said, there are a few things I am a bit worried about. It seems like to sample the ELBO there is a need to resample all the chains ?

---

> ### Author Response · Authors · 2022-08-02
> **Response to Reviewer xhuG**
>
> Thank you for your insightful comments. We answer your questions to address your concern.
>
> > It seems like to sample the ELBO there is a need to resample all the chains ?
>
> Is this about the ELBO calculation in Eq. (20) for the evaluation of LAEs? If so, resampling the encoder's parameters (i.e., $\Phi$ and $\psi$) at test time is not required, because we fix those parameters after training.
>
> > You have used EM numerical solver, which has a lof of well defined numerical stability criterion, for example : Higham, D.J., 2001. An algorithmic introduction to numerical simulation of stochastic differential equations. SIAM review, 43(3), pp.525-546. Did you check any of them ?
>
> Although we did not check such criterions, we instead use Metropolis Hastings (MH) rejection steps to modify the discretization error caused by EM numerical solver as described in 3.1.
> The introduction of MH rejection steps guarantees that the generated Markov chains are ergodic and have the target true posterior as its stationary distribution even when the EM solver is nonequilibrium due to discretization (see e.g., [1, 2]).
> In fact, we observed that MH rejection steps are important to stabilize the training of LAEs as described in Section 5.2.
>
> > What is really fascinating about Langevin dynamics is that for a certain class you derive a shadow lemma which means you are tractable towards your “real” underlying distribution for example : https://arxiv.org/abs/1107.2967 Can this analysis somehow improve the theorem on G
>
> Thank you for introducing us to the important related work. We think that such analysis on nonequilibrium Langevin dynamics simulators is useful to improve our theorem so that it can consider the discretization error caused by numerical solvers. Though our theorem verifies that the stochastic differential equation induced by the encoder has the underlying target distribution as its equilibrium, it does not care about the discretization error of numerical solvers (like EM solver). In our practical implementation, we fix the discretization error by using MH rejection as mentioned above, but it is possible to use other methods such as the one proposed in the paper you mentioned.
>
> We would be glad to respond to any further questions and comments that you may have.
>
> Thanks.
>
> **References**
>
> [1] Gareth Roberts and Jeffrey Rosenthal. Optimal Scaling of Discrete Approximations to Langevin Diffusions. Journal of the Royal Statistical Society: Series B (Statistical Methodology), 60: 255-268, 1998. https://doi.org/10.1111/1467-9868.00123
>
> [2] T. Xifara et al. Langevin diffusions and the Metropolis-adjusted Langevin algorithm. arXiv preprint arXiv:1309.2983, 2013. https://arxiv.org/abs/1309.2983

---

> > ### Comment · Reviewer_xhuG · 2022-08-09
> > **Thank you for the in depth response**
> >
> > > Is this about the ELBO calculation in Eq. (20) for the evaluation of LAEs? If so, resampling the encoder's parameters (i.e.,  and ) at test time is not required, because we fix those parameters after training.
> >
> > Thank you this was not clear to me that you hold the parameters fixed after training. Are you still robust to distributional shifts ?
> >
> > > Although we did not check such criterions, we instead use Metropolis Hastings (MH) rejection steps to modify the discretization error caused by EM numerical solver as described in 3.1. The introduction of MH rejection steps guarantees that the generated Markov chains are ergodic and have the target true posterior as its stationary distribution even when the EM solver is nonequilibrium due to discretization (see e.g., [1, 2]). In fact, we observed that MH rejection steps are important to stabilize the training of LAEs as described in Section 5.2.
> >
> > That is a good point, still it would have been nice to understand how does the MH rejections steps effect the EM error bounds
> >
> > > Thank you for introducing us to the important related work. We think that such analysis on nonequilibrium Langevin dynamics simulators is useful to improve our theorem so that it can consider the discretization error caused by numerical solvers. Though our theorem verifies that the stochastic differential equation induced by the encoder has the underlying target distribution as its equilibrium, it does not care about the discretization error of numerical solvers (like EM solver). In our practical implementation, we fix the discretization error by using MH rejection as mentioned above, but it is possible to use other methods such as the one proposed in the paper you mentioned.
> >
> > Notes, again, I feel like there are a lot of theoretical missing pieces to this.
> > For now I stand by my score

---

### Meta-Review · Area_Chair_QVro · 2022-08-28

**Recommendation:** Accept
**Confidence:** Certain

**Metareview:**

The problem addressed in this paper is the one of inference in deep generative latent variable model. The paper proposes a novel approach with an amortized approximation to the joint distribution of data and latent.

All three reviewers liked the paper with one reviewer being a bit concerned about the soundness of the MCMC convergence proof.  In this meta-reviewers' opinion the algorithm (Alg 2) is sound although there might be subtleties here because it mixes posterior parameter updates with generative model parameter updates and latent samples.

The paper is original, clear and numerical evaluations sufficient so acceptance is recommended.

**Award:**

No

---

### Decision · Program_Chairs · 2022-09-14

Accept